# Physical and Morphological Properties of Tough and Transparent PMMA-Based Blends Modified with Polyrotaxane

**DOI:** 10.3390/polym12081790

**Published:** 2020-08-10

**Authors:** Akira Ishigami, Kazuki Watanabe, Takashi Kurose, Hiroshi Ito

**Affiliations:** 1Graduate School of Organic Materials Science, Yamagata University, 4-3-16 Jonan, Yonezawa, Yamagata 992-8510, Japan; akira.ishigami@yz.yamagata-u.ac.jp (A.I.); tye21943@st.yamagata-u.ac.jp (K.W.); 2Research Center for Green Materials and Advanced Processing, Yamagata University, 4-3-16 Jonan, Yonezawa, Yamagata 992-8510, Japan; takashi.kurose@yz.yamagata-u.ac.jp

**Keywords:** polymer blends, polymer alloys, polyrotaxane, toughening, reactive blend

## Abstract

We prepared several novel, tough, and transparent poly(methyl methacrylate) (PMMA) blends modified with polyrotaxane (PR) and evaluated their physical properties and morphologies. A styrene/methyl methacrylate/maleic anhydride (SMM) copolymer that was miscible with PMMA was used as a reactive compatibilizer to enhance interfacial adhesion between the matrix resin and PR. A twin-screw melt-kneading extruder was used to prepare the polymer blends, and their thermal, morphological, optical, and mechanical properties were characterized. The effect of PR was evaluated by analyzing the deformation behavior of the blends in notched three-point bending tests. A PMMA/PR blend was immiscible and appeared to be a phase-separated system. However, when SMM was added as a compatibilizer, PR was partially miscible and did not form observable PR domains. Viscosity increased, and the glass transition temperature (*T*_g_) of the matrix resin decreased. The surface hardness of a PMMA/SMM/PR blend was only 15% lower than that of PMMA. A 2.5-fold increase in elongation at breakage was observed, and the tensile strength and Young’s modulus decreased by 16%. The PMMA/SMM/PR blend had 60% higher impact strength than PMMA in notched Charpy impact test, which indicated that the balance between stiffness and ductility was excellent. PR served as a starting point for plastic deformation in the PMMA/SMM/PR blend. We found that PR could initiate void and craze formation, even when it was finely dispersed at the nanoscale. The stress-relieving effect of PR was effective when it was tightly bound at the interfaces. The materials obtained in this study are expected to make a significant contribution to reducing the weight of the products by applying them as a replacement for glass.

## 1. Introduction

Poly(methyl methacrylate) (PMMA) is a particularly hard and transparent amorphous thermoplastic polymer that is commonly used as an alternative to glass in optical product manufacturing. PMMA is also highly weather resistant, so it is widely utilized as a general purpose resin to coat or laminate steel and resin plates for outdoor applications [1]. However, the impact resistance and ductility of PMMA are quite poor [2]. Denaturing PMMA can impart impact resistance and ductility. Denaturation methods involve blending PMMA with rubber or an elastomer to produce rubber-reinforced PMMA (RTPMMA). Laatsch et al. transitioned PMMA from the brittle deformation mode to the ductile deformation mode by dispersing core–shell particles in a PMMA matrix [3]. Li et al. successfully fabricated transparent and highly ductile PC/PMMA blends by using a high shear field [4]. The particles were 180 nm in diameter and contained cores composed of rubber components, while PMMA comprised the shells. Wang et al. fabricated porous nanocellular PMMA/thermoplastic polyurethanes (TPU) that significantly improved toughness and promoted molecular chain movement, slippage, and nanofibrilization [5]. Mao et al. blended chlorinated polyethylene (40 wt %) and PMMA to form a co-continuous structure that increased impact resistance 26-fold relative to that of PMMA [6]. Blending PMMA with 20–30 wt % of a rubber or elastomeric component can provide sufficient impact resistance and ductility, and the stiffness of the resulting PMMA-based polymer blends is significantly lower [3,4,5,6,7,8,9,10,11]. Amorphous polymeric materials must be highly transparent, which significantly limits the range of modification methods that can be used. This is because it is necessary to either optimize the refractive index of the modifier or use small particles that do not scatter light [3,12,13,14]. Designing materials to modify the properties of amorphous polymer materials is thus considerably more difficult than it is for crystalline polymers, and alternatives to the RTPMMA property modification approach are urgently needed.

Polyrotaxane (PR) has recently attracted interest as part of a new approach to modifying polymeric materials. Harada et al. synthesized a polyrotaxane supermolecule by threading polyethylene glycol (PEG), a linear polymer, through cyclodextrin (CD) rings [15]. Both ends of each PEG molecule were capped with adamantanamine to prevent the CD rings from escaping. Ito and Takeoka et al. developed a novel elastomer by grafting poly(ε-caprolactone) (PCL) onto the CD rings of a cyclic PR molecule. They reported that the elongation rate was significantly better than those of conventional elastomers [16,17,18]. This was attributed to the CD sliding ring mechanism, which enabled the crosslinking points to move freely. It was also reported that reactive blending of poly(lactic acid) (PLA) with PR and methylene diphenyl diisocyanate produced stress concentration centers that absorbed a large amount of energy during impact and tension processes [19,20]. PR has been applied to soft materials, including gels and elastomers, but its application to hard materials and the associated mechanisms have not been sufficiently investigated.

In this study, PR modification was employed to fabricate a novel PMMA-based polymer blend that exhibited excellent toughness and transparency. We used a reactive styrene/methyl methacrylate/maleic anhydride (SMM) copolymer as a compatibilizer to enhance interfacial adhesion between PR and the matrix resin. SMM is a transparent and amorphous thermoplastic resin that is miscible with PMMA [21,22]. An ester bond is formed between the terminal hydroxyl group of the PCL chain attached to the CD of PR and the maleic anhydride group of SMM. Therefore, it is thought that they make the formation of the PR–SMM copolymer and improve its compatibility. In this study, polymer blends were prepared using a continuous, twin-screw melt-kneading extrusion process. Injection molding was performed to prepare test specimens to evaluate the mechanical properties of the blends. We also evaluated the physical properties of the products and compared them to those of materials that were manufactured through a wide range of industrial methods. We characterized the thermal, morphological, optical, and mechanical properties of the fabricated polymer blends, and the function of PR was evaluated by analyzing deformation behavior in notched three-point bending tests.

## 2. Experimental

### 2.1. Materials and Methods

PMMA (VH001, *M*_w_ 11 × 10^4^) was obtained from Mitsubishi Chemical Corp. (Tokyo, Japan) and used as a matrix resin for the polymer blends. PR (SH3400, *M*_w_ 70 × 10^4^) was purchased from ASM Inc. (Chiba, Japan) and used to modify the properties of the matrix resin. The chemical structure of PR is illustrated in Figure 1. SMM (R200, *M*_w_ 17 × 10^4^) was obtained from Denka Co., Ltd., (Tokyo, Japan) and blended to enhance interfacial adhesion between PR and the matrix resin. The PMMA and SMM pellets were dried at 80 °C for 12 h prior to use, and PR was dried in vacuo at room temperature for at least 24 h prior to use.

### 2.2. Sample Preparation

A KZW15TW-30MG-NH (−700) twin-screw melt-kneading extruder (Technovel Corp., Osaka, Japan) with a screw length/screw diameter (*L/D*) ratio of 45 (screw diameter *D*:15-mm-Φ) was used to prepare the polymer blends at 230 °C and a screw rotation rate of 100 rpm. The samples were coded as follows according to the type of raw material and the ratio of the ingredients: PMMA (neat PMMA), SMM (neat SMM), PMSM (PMMA/SMM = 79/21 wt %), PR5 (PMMA/PR = 95/5 wt %), PR10 (PMMA/PR = 90/10 wt %), PSR5 (PMMA/SMM/PR = 75/20/5 wt %), and PSR10 (PMMA/SMM/PR = 70/20/10 wt %). The formulation ratio of each sample is shown in Table 1.

Dumbbell-shaped specimens (JIS standard: Type K-7162-1BB, parallel section width: 2 mm, parallel section thickness: 2 mm, specimen length: 50 mm) of the pelletized polymer blend materials were prepared using an EP5 small injection-molding machine (Nissei Plastic Industrial Co., Ltd., Nagano, Japan) for tensile tests and notched Charpy impact test. The injection and mold temperatures were 240 and 80 °C, respectively, and a mold clamping force of 5 t was applied. Three-point bending test specimens (width: 12.5 mm, length: 50.0 mm, thickness: 6.35 mm) with notch depths of 2 mm were prepared in 80 °C molds using an FNX80 large injection-molding machine (Nissei Plastic Industrial Co., Ltd., Nagano, Japan). The injection temperature was 240 °C, and a mold clamping force of 80 t was applied. The test specimens were used for notched three-point bending tests and deformation analysis. Test specimens for transmittance measurements were molded using an 11FD press (Imoto Machinery Co., Ltd., Kyoto, Japan). Flat specimens with thicknesses of 300 μm were pressed for 2 min at 240 °C under 10 MPa of pressure. In Section 3, the values reported for each sample are the average values obtained from the five specimens.

### 2.3. Characterization

#### 2.3.1. Differential Scanning Calorimetry (DSC)

A Q200 DSC (TA Instruments Japan Inc., Tokyo, Japan) was used to analyze the thermal properties of the neat matrix resin and the polymer blends. Approximately 5 mg was collected from the injection-molded specimens and used for DSC measurement. In addition, the first heating scan thermogram was employed. Thermograms were recorded in the temperature range from 0 to 200 °C as the samples were heated at a rate of 3 °C/min. All samples were maintained in a nitrogen atmosphere for the DSC measurements.

#### 2.3.2. Melt Flow Rate (MFR) Measurements

The MFRs of the neat matrix resin and the polymer blends were measured to evaluate their flowability using a F-F01 melt indexer (Toyo Seiki Seisaku-Sho Ltd., Tokyo, Japan). The MFRs were obtained by measuring the amount of resin extruded over 10 min from a die installed at the bottom of a 230 °C cylinder under a 5.0 kg load.

#### 2.3.3. Transmission Electron Microscopy (TEM)

The morphologies of the PMMA/PR and PMMA/SMM/PR blends were characterized using a JEM-2100F TEM (JEOL Ltd., Tokyo, Japan). The PR dispersed throughout the matrix resin was stained with ruthenium tetroxide (RuO_4_). Ultrathin (100 nm) sections were then obtained using an Ultracut UCT ultramicrotome (Leica Microsystems, Inc., Tokyo, Japan), and TEM images were collected at an accelerating voltage of 200 kV.

#### 2.3.4. Transmittance Evaluation

Transmittance by the press-molded samples was measured at wavelengths in the range from 380 to 780 nm using an angle-dependent total reflection measurement unit (Lambda Vision Inc., Kanagawa, Japan) equipped with a D65 light source to evaluate transparency.

#### 2.3.5. Mechanical Properties

##### Surface Hardness Measurements

The surface hardness of each dumbbell-shaped specimen was measured using a G200 nanoindenter (MTS Nano Instruments Inc., Oak Ridge, TN, USA). The Berkovitch indenter had a triangular pyramidal shape, and hardness was determined by performing continuous stiffness measurements up to an indentation depth of 10 µm [23]. The reported surface hardness value was averaged over the indentation range from 500 to 1900 nm to eliminate the influences of surface roughness and indentation size.

##### Tensile Strength

The injection-molded products were subjected to tensile testing at a crosshead speed of 5 mm/min using a universal testing machine (Strograph VGS-E01, Toyo Seiki Seisaku-Sho Ltd., Tokyo, Japan). The chuck distance was 22 mm, and the value of strain was calculated from the displacement with respect to the initial chuck distance.

##### Impact Strength

Notched Charpy impact tests (edgewise) of the injection-molded samples were performed, where the lifting angle of the hammer was 90°, the moment of the hammer was 0.1 N·m, the hammer speed was 2.1 m/s, and the hammer energy was 0.1 J, using a digital impact tester (DG-1B: Toyo Seiki Seisaku-Sho Ltd., Tokyo, Japan). The notch with a depth of 0.3 mm was applied to the injection-molded products using an automatic notching machine (A-4E, Toyo Seiki Seisaku-Sho Ltd., Tokyo, Japan). In this Charpy impact test, evaluation was performed as a relative comparison between samples.

#### 2.3.6. Analysis of Deformation Behavior

U-notched test specimens were subjected to three-point bending tests using a universal testing machine (Autograph AG-5kNE, Shimadzu Corporation, Kyoto, Japan). Testing was performed at a rate of 2 mm/min over a span of 40 mm. The bending tests were stopped at the point of strain immediately before rupture, and the specimens were fixed with epoxy. The U-notch regions of the fixed specimens were removed in 100 µm thick sections using a microtome (Leica Microsystems, Inc., Tokyo, Japan). The regions of plastic deformation in the thin sections were examined using a polarized light microscope (BX51, Olympus Corp., Tokyo, Japan). The specimens subjected to bending tests were immersed in a liquid nitrogen bath for 5 min. Immediately after removal from the bath, each specimen was fractured in the direction perpendicular to the notch. The fractured surfaces were analyzed by collecting scanning electron microscope (SEM) images (Miniscope TM3030Plus, Hitachi High-Tech Corporation, Tokyo, Japan).

## 3. Results and Discussion

### 3.1. Thermal Characterization

The *T*_g_ and *T*_m_ values of the injection-molded samples determined from the DSC measurements are shown in Figure 2 and Table 1. The flowability results obtained by performing MFR measurements are also shown in Table 1. The *T*_g_ of PMSM was consistent with the calculated *T*_g_ of 117.6 °C obtained using the Fox formula [24], which suggested that PMMA and SMM were compatible at the molecular level. The *T*_g_’s of PR5 and PR10 were quite similar to the *T*_g_ of PMMA, which suggested that the blended PR phase separated from the PMMA matrix resin. The PR10 sample had a particularly high PR content, and its *T*_m_ was near the melting point of neat PR. This indicated that PR separated from the PMMA matrix resin as a crystalline phase. The MFRs of PR5 and PR10 were higher than that of PMMA due to increased fluidity, which was a consequence of blending PRs with lower melting points. The *T*_g_’s of PSR5 and PSR10 blended with SMM, which contained maleic anhydride groups, were shifted to lower temperatures relative to the *T*_g_ of the PMSM matrix resin. The *T*_g_’s of PSR5 and PSR10 were 4.9 and 7.3 °C lower, respectively, than the *T*_g_ of PMSM. The PR-induced melting peak was absent in the thermogram of PSR10. The MFRs of PSR5 and PSR10 were significantly lower than that of PMSM. The viscosity was thought to have increased with the increase in molecular weight due to chemical bonding during the melt-kneading process. The dispersion of PR and its interfacial adhesion with the matrix resin were also expected to improve.

### 3.2. Transparency and Morphological Properties

The transmittance spectra of the press-molded products are shown in Figure 3. PMMA, SMM, and PMSM had transmittance values exceeding 90% over the entire visible light range. PSR5 and PSR10 also had high transmittance values of ~90%. However, they absorbed light at wavelengths below 500 nm. This suggested that the systems were not perfectly miscible and that light was scattered by minute phase-separated structures. In contrast, PR5 and PR10 had transmittance values below 75%. This was due to the immiscibility of PR and the matrix resin and the presence of PR as a large domain.

TEM images of the PMMA/PR and PMMA/SMM/PR blends prepared by the melt-kneading process are shown in Figure 4. The RuO_4_ stain made the PR domains in the images of PR5 in Figure 4A-1,A-2 and of PR10 in Figure 4B-1,B-2 appear black. The PR domains dispersed in the PMMA matrix had diameters ranging from 100 to 200 nm. Like the results of thermal characterization, the TEM images indicated that PR formed phase-separated structures. The domains in PR10 varied greatly in size. In contrast, no clear phase separation was observed in PSR5 (Figure 4C-1,C-2) and PSR10 (Figure 4D-1,D-2). A reaction between SMM and PR in PSR5 and PSR10 was induced by melting and kneading. These results suggested that the reaction between PR and SMM promoted fine dispersion. Unlike the PMMA/PR blends, the PMSM/PR blends exhibited gradients rather than well-defined interfaces, which indicated that PR was partially miscible with the matrix resin. PSR5 and PSR10 were not fully miscible with each other, but they transmitted well. This was because phase separation was inhibited by the reaction between SMM and PR.

### 3.3. Evaluation of Mechanical Properties

#### 3.3.1. Surface Hardness

The surface hardness values for each sample determined via continuous stiffness measurements are shown in Figure 5. The surface hardness of PMMA and SMM was characteristically high due to their rigid molecular skeletons. The surface hardness of PMSM, which was a miscible system, was comparable to that of PMMA. The surface hardness of PR5 and PR10 was lower due to the presence of flexible PR domains with diameters ranging from 100 to 200 nm. The surface hardness of PR10 was 26% lower than that of PMMA, while the surface hardness values of PSR5 and PSR10 were only 15% lower than those of PMMA and PSMS, respectively. The surface hardness of PSR10 was particularly high, even though it contained 10 wt % PR. Based on the internal structure observed in the TEM images, PR that reacted with SMM was not present in a clearly separate domain. The PR was finely dispersed, so the high surface hardness of the matrix resin could be retained without local decreases in hardness. The hardness of the blended materials was generally lower due to the presence of flexible materials [25]. However, we concluded that by limiting the size of the partially miscible domains to the nanoscale, we could obtain a structure that retained the high surface hardness of PMMA.

#### 3.3.2. Tensile Tests

Stress–strain curves obtained from the tensile tests are shown in Figure 6. The tensile properties determined from the stress–strain curves are shown in Table 2. The Young’s moduli of PMMA, SMM, and PMSM were high, but the samples were brittle and broke in the midst of yielding. Notably, the SMM sample broke at just 4% strain without yielding. Blending PR with PMMA increased the rupture strain of PR5 to 21.3%, but PR10 ruptured at 13.0% strain. Morphological analysis of PR10 showed that the diameters of the PR domains dispersed in the matrix resin varied widely. The non-uniform distribution of the PR domains made the concentration of localized stress more likely, so the sample was easily broken. The interfaces between PR and the matrix resin in PSR5 and PSR10 were controlled. Each of these samples exhibited a yield point and a large increase in strain at the time of breakage, which indicated a strong modifying effect. The breaking strain of PSR10 was approximately 2.5 times higher than that of PMMA, and its breaking strain energy was approximately 2.3 times higher. The tensile strength and Young’s modulus of PSR10 were only 16% lower than those of PMMA, which indicated that the stiffness and ductility of the blended material were balanced well.

#### 3.3.3. Notched Charpy Impact Tests

The results of the notched Charpy impact tests are shown in Figure 7. As expected, PMMA, SMM, and PMSM displayed quite low impact resistance in the notched Charpy impact test at impact energies as low as ~3 kJ/m^2^. The impact energies of PR5 and PR10 were lower than those of the matrix resins, which did not appear to have significant modifying effects on physical properties based on the tensile tests. On the other hand, the impact energies of PSR5 and PSR10 were approximately 60% higher than that of PMMA. In PR5 and PR10 without SMM, the modification effect by PR could not be confirmed. However, controlling interfacial adhesion between the matrix resin and PR had an increase in effectiveness on impact strength. This suggested that partially miscible PR relieved stress, even during rapid deformation.

### 3.4. Analysis of Deformation Behavior

U-notched specimens were subjected to three-point bending tests in which stress was concentrated at the tips of the notches. Strain at the notch tips occurred in the vertical direction relative to displacement. Poisson contraction was thus suppressed at the notch tip, which resulted in the constraint of the strain. As a result, deformation occurred solely through uniaxial elongation [26,27,28]. This test method is known to be effective for assessing the toughness of polymer blends [6,29,30,31]. The toughness values of the U-notched matrix resin specimens, the PMMA/PR blends, and the PMMA/SMM/PR blends determined through three-point bending analysis are shown in Figure 8. Displacement in each sample at breakage correlated with elongation at breakage in the tensile tests, and breakage displacement was no better in the PMMA/PR blended system. On the other hand, improved interfacial adhesion in the PMMA/SMM/PR blended system enhanced breakage displacement. This was particularly evident in PSR10, which had the most significant breakage displacement. This result suggested that PR had a functional modifying effect.

To confirm the deformation behavior of each specimen during the notched three-point bending test, the test was stopped at the point at which displacement immediately preceded breakage, and the specimens were fixed with epoxy. Polarization microscope images of notched tips from PMMA, PR10, and PSR10 are shown in Figure 9A–C. No changes were observed in the notched tips of PMMA and PR10, even immediately prior to breakage. The stress relaxation mechanisms of void and craze formation did not occur. Therefore, the extension of locally occurring cracks was assumed to occur at the same time, which lead to brittle fracture. A large, dark plastic deformation region was observed at the tip of the notch in PSR10. In other words, light-scattering structures were thought to have formed in PSR10. SEM images of the surfaces of freeze-fractured specimens are shown in Figure 9a–c. The fractured surfaces of PMMA (Figure 9a) and PR10 (Figure 9b) were smooth. PR-induced voids where craze would originate were absent in PR10, which resulted in facile exfoliation at the fragile interface between the matrix resin and PR. The rapid growth of cracks resulted in fracture because they were unable to form voids that would inhibit the expansion of craze. In contrast, craze formation near voids was confirmed in PSR10 (Figure 9c). The PR domains in PSR10 were finely dispersed. When stress was applied, the PR domains became stress concentration points and generated craze at the interface. Craze expanded to the surrounding PR domains, but the tightly bound and partially miscible interfaces inhibited its growth. Significant crazing near the dispersed PR domains grew into the voids, which afforded a large energy absorption capacity.

## 4. Conclusions

In this study, PMMA, SMM, and PR polymer blends were prepared using a twin-screw melt-kneading extruder. SMM reacted with hydroxy groups in PR. The matrix resin and polymer blends were molded into test samples for thermal and mechanical characterization using an injection-molding machine. The internal structures and deformation behaviors of the blends were also evaluated. A PMMA/PR blend formed a phase-separated structure in which the interface and PR domains did not interact. As a result, the PMMA/PR blend transmitted less visible light than PMMA. SMM was added to obtain a PMMA/SMM/PR blend with a higher molecular weight and greater viscosity, which was due to the reaction of PR with SMM. The absence of distinct PR domains in the TEM images and a decrease in *T*_g_ detected via DSC indicated that the PR domain interfaces were partially miscible after PR reacted with SMM. As a result, the PMMA/SMM/PR blend transmitted ~90% of visible light. Controlling the interfaces between the matrix resin and PR also had a positive effect on mechanical properties. The surface hardness of PMMA/PR blends was approximately 26% lower than that of PMMA due to the presence of large phase-separated PR domains. However, the surface hardness of the PMMA/SMM/PR blend was only 15% lower than that of PMMA. PR was partially miscible after reacting with SMM, and its dispersion could be controlled at the nanoscale. This enabled us to generate a structure with high surface hardness. The breaking strain of the PMMA/SMM/PR blend in tensile tests was ~2.5 times higher than the breaking strain of PMMA. The breaking strain energy was approximately 2.3 times higher, while the tensile strength and Young’s modulus of the blend were 16% lower. In addition, the impact energies of PMMA/SMM/PR blends were approximately 60% higher than that of PMMA. It could thus be concluded that the balance between the stiffness and ductility of the blend was excellent. No plastic deformation region was observed in PMMA, and the stress relaxation mechanism was only functional immediately prior to breakage. In contrast, the PMMA/SMM/PR blend contained large plastic deformation regions that were formed by PR-initiated crazing and voids. We found that PR caused void and craze formation, even when it was finely dispersed in a transparent material. We also found that PR had a stress-relieving effect when it was tightly bound at the interfaces.

## Figures and Tables

**Figure 1 polymers-12-01790-f001:**
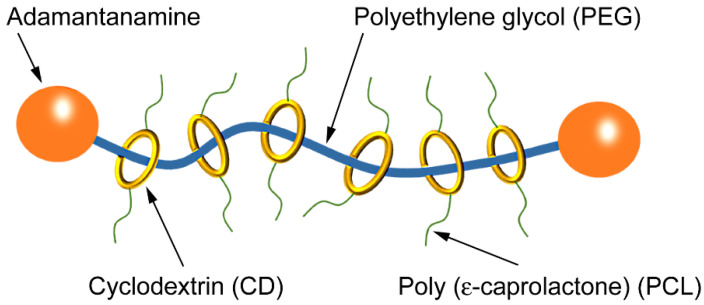
The chemical structure of polyrotaxane (PR).

**Figure 2 polymers-12-01790-f002:**
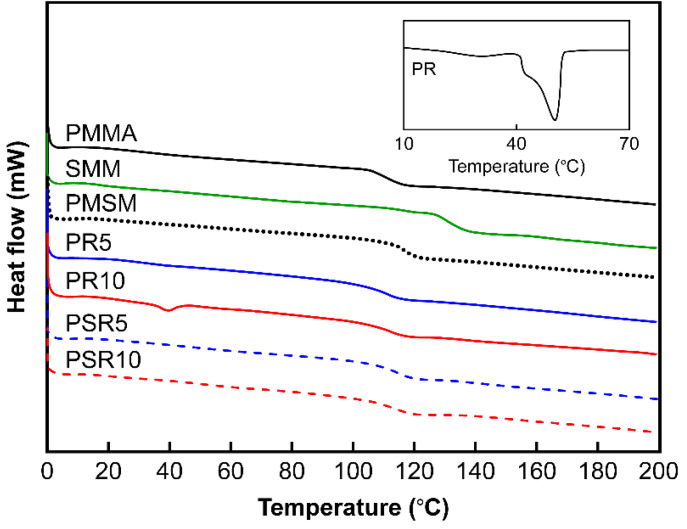
DSC thermograms of the matrix resins, PR, PMMA/PR blends, and PMMA/SMM/PR blends.

**Figure 3 polymers-12-01790-f003:**
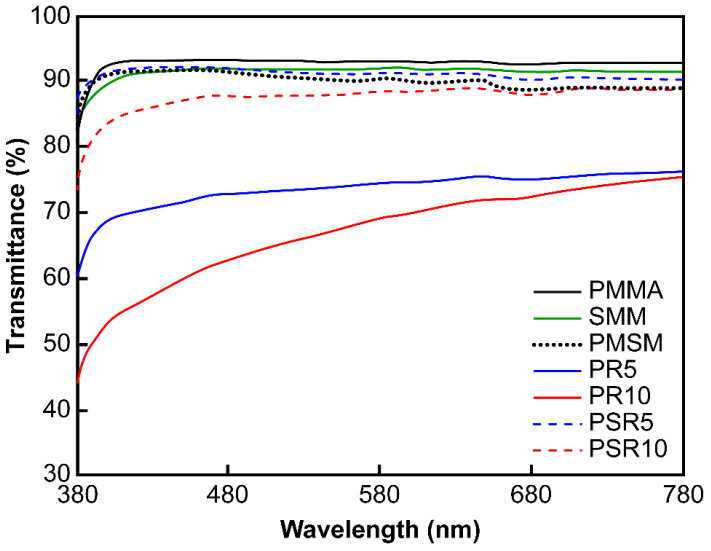
Transmittance spectra of the matrix resins, PMMA/PR blends, and the PMMA/SMM/PR blends.

**Figure 4 polymers-12-01790-f004:**
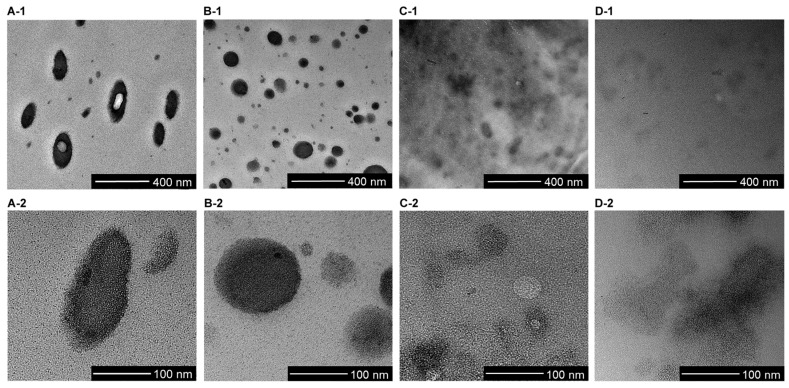
TEM images showing the morphological structures of the (**A-1**,**A-2**) PR5, (**B-1**,**B-2**) PR10, (**C-1**,**C-2**) PSR5, and (**D-1**,**D-2**) PSR10 blends. The dark areas are PR phases stained with RuO_4_.

**Figure 5 polymers-12-01790-f005:**
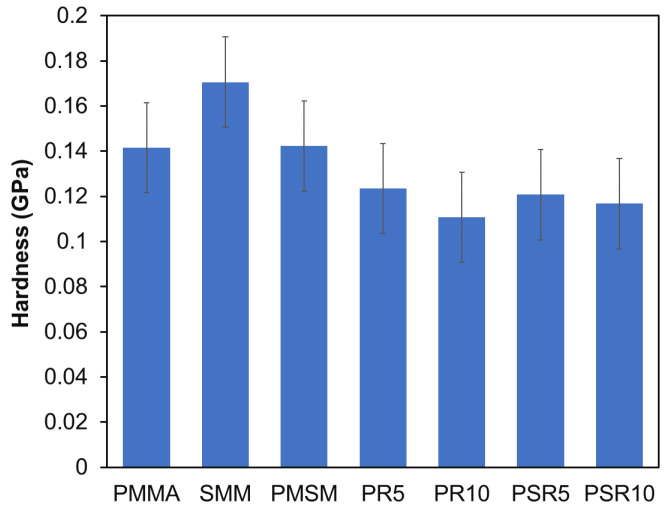
Hardness of the pressed matrix resins, PMMA/PR blends, and PMMA/SMM/PR blends.

**Figure 6 polymers-12-01790-f006:**
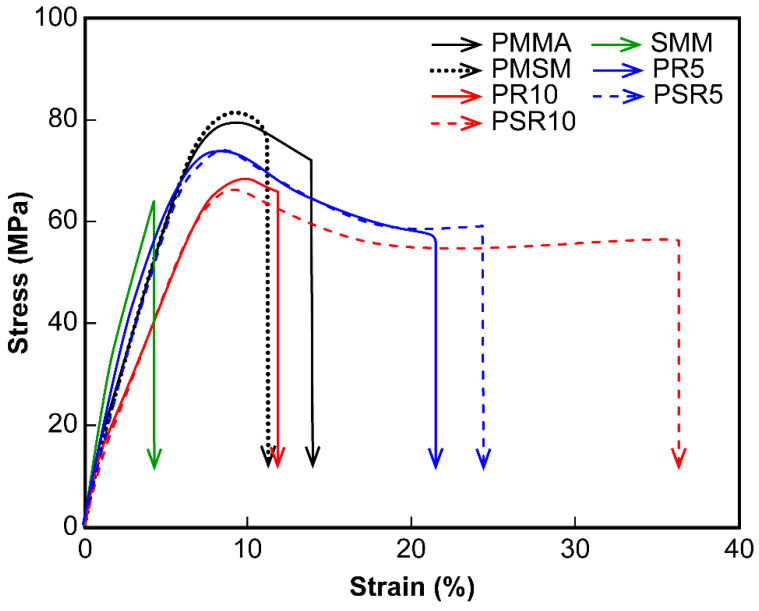
Stress–strain curves of the matrix resins, PMMA/PR blends, and PMMA/SMM/PR blends.

**Figure 7 polymers-12-01790-f007:**
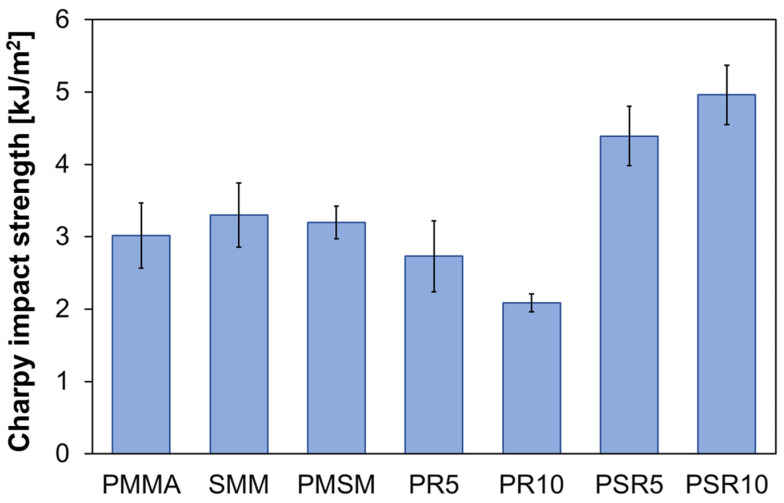
Notched Charpy impact test results for the matrix resins, PMMA/PR blends, and PMMA/SMM/PR blends.

**Figure 8 polymers-12-01790-f008:**
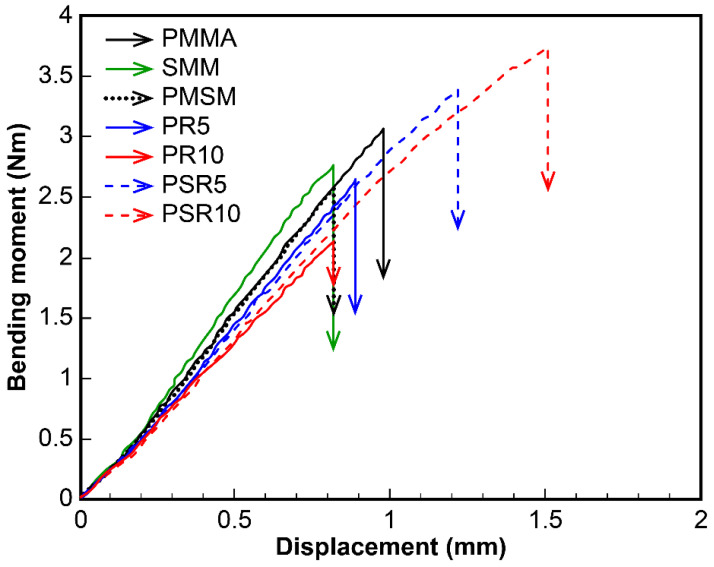
Toughness of U-notched PMMA, PMMA/PR blend, and PMMA/SMM/PR blend specimens measured in three-point bending tests.

**Figure 9 polymers-12-01790-f009:**
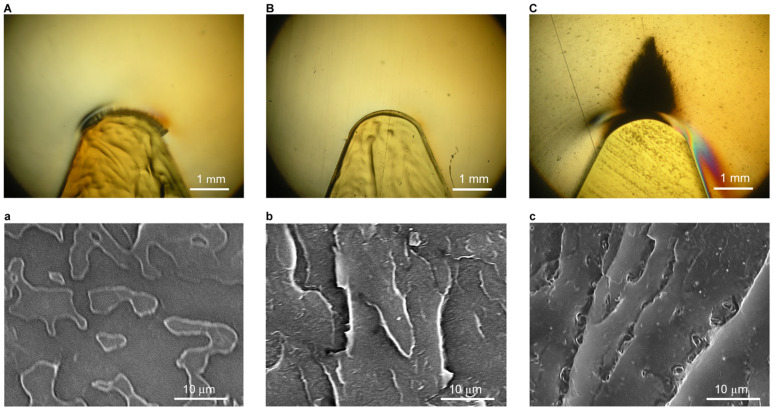
Polarized light micrographs of (**A**) PMMA, (**B**) PR10, and (**C**) PSR10. SEM images of (**a**) PMMA, (**b**) PR10, and (**c**) PSR10 in regions of plastic deformation at the notch tips.

**Table 1 polymers-12-01790-t001:** *T*_m_, *T*_g_, and melt flow rate (MFR) values of the matrix resins; neat PR; blends of poly(methyl methacrylate) (PMMA) and PR; and blends of PMMA, styrene/methyl methacrylate/maleic anhydride (SMM), and PR.

Code	PMMA/SMM/PR (wt %)	*T*_m_ (°C)	*T*_g_ (°C)	MFR (g/10 min)
PMMA	100/0/0	-	114.1	2.5
SMM	0/100/0	-	132.7	4.2
PMSM	79/21/0	-	118.6	2.9
PR	0/0/100	50.0	-	-
PR5	95/0/5	-	112.0	3.9
PR10	90/0/10	39.7	111.9	5.0
PSR5	75/20/5	-	113.7	0.6
PSR10	70/20/10	-	111.3	0.2

**Table 2 polymers-12-01790-t002:** Mechanical properties of the matrix resins, PMMA/PR blends, and PMMA/SMM/PR blends.

Code	Tensile Strength (MPa)	Elongation at Break (%)	Young’s Modulus (MPa)	Breaking Energy (J/m^3^)
PMMA	79.7	14.8	2296	8.3
SMM	63.8	4.0	2557	1.6
PMSM	81.5	11.6	2290	6.3
PR5	74.3	21.3	2188	12.6
PR10	68.4	13.0	2069	5.6
PSR5	74.3	24.4	2024	14.1
PSR10	67.0	37.6	1916	19.3

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
