# Peer review of "Physical and Morphological Properties of Tough and Transparent PMMA-Based Blends Modified with Polyrotaxane"

_polymers, 2020, doi:10.3390/polym12081790_

Round 1

Reviewer 1 Report

The manuscript by Ito et al reported the transparent PMMA/polyrotaxane blends b reactive compatibilization. The physical properties and the morphologies have been investigated. The results are interesting and the manuscript is well arranged. This reviewer suggests the publication after the following minor revisions: 1) The authors should comment on the reactive compatibilization mechanism or give the scheme of the reactive compatibilization. 2) The detail PR structural parameters should be given in the experimental section. 3) Some transparent PMMA alloys should be mentioned in the introduction part. For example: Polym Eng. Sci. 2011,51,1437 etc.

Author Response

Thank you very much for your valuable comments in our manuscripts. We modified and revised this manuscript under your comments.

The revised manuscript and comments on the revisions are attached.

Reviewer 2 Report

The paper titled “ Physical and morphological properties of tough and transparent PMMA-based blends modified with polyrotaxane” appears as very interesting, showing as the main novelty the use of polyrotaxane as a polymer capable of improving the mechanical properties of the PMMA without much loss of transparency. Since this viewpoint the work is attractive and deserves being published.

However, a more profound reading of the text makes emerges a series of inconsistencies, mainly related to the experimental section which obliges recommending a profound revision of the manuscript.

Without being exhaustive, below these lines a series of comments are pointed out, and this reviewer recommends the authors to consider them by providing more information (when required), or by answering this reviewer concerns in a convincing manner.

  • Please, provide some properties of the polymers and the compatibilizer used and not just the supplier. This is important for traceability and easier comprehension of the further discussed.
  • In the DSC study, please, provide the amount of sample used and the number of scans.
  • Concerning also to the DSC study, the authors have performed a single heating scan. Any reason for it? Note that this scan incorporates the information of both the processing step and of the material itself. So, this reviewer wonders if the authors have considered the latter. Since the processing conditions used are the same for all the samples in spite of the very different nature of the compounds (the authors used the same condition for processing so different neat materials as PMMA and SMM, and all the other compounds), and the effect of this may play a very different role depending on the concentration. Note that when a material is tested, the information obtained id a mixture of that provided by the processing operations plus the one due to the structure of the material.  So, this reviewer wonders why the authors have not performed a cooling and a further second heating scan in order of erasing the previous thermal history of the material, and then isolating the thermal behavior of the material itself.
  • Is there any reason for using TEM in absence of other observations at lower magnification levels? Since microscopy follows a hierarchical scheme, the logical is to observe from eye to TEM by using lower resolution techniques providing a more accurate panorama about the mixing and dispersion of the phases. This is critical since the authors recognize in the discussion section that some differences in properties may be due to a bad mixing, and so to non homogeneous samples.
  • In the Tensile section, the authors avoid to provide information about the use or not of extensometer. The gap distance is also needed. This is important to make opinion about the precision of the results reported.
  • In the Impact section, this reviewer wonders if the authors have used the dumb-bell shaped specimens. This is not clear in the text. Additionally, the authors must inform if the samples have been flat-wise of edgewise used. The information about the champions for the impact testing must be better defined (for traceability sake). Also in this section, it must be remarked that just the mass of the hummer is not sufficient to define the test conditions by itself. At least, it is necessary to inform about the energy provided (dependent of the hammer angle) and the testing rate. In absence of these (jointly to the information about the specimens), the information is incomplete as to replicate the experiments. Please, provide the number of samples measured.
  • The figure 5 needs error bars. Please, provide them. Based in my own experience on this type of tests. The error bars for the PR5, PR10, PSR5 and PSR6 may surely overlap, and so, any discussion on these results must be necessarily spurious.
  • The Figure 7 includes error bars for impact parameter, but in this occasion, the error bars clearly overlap. Please, a discussion for this is needed. Nevertheless, in absence of the test conditions, sample better specifications and number of samples tested, the results hardly may be robust for a reader.
  • The authors use the MFR to suggest that the reaction between SMM and PR may be taken place: The MFRs of PSR5 and PSR10 were significantly lower than that of PMSM, which suggested that the maleic anhydride groups in SMM reacted with hydroxy groups in PR to form chemical bonds between SMM and PR.”. Note that this is a bold suggestion in absence of more evidences. The sensibility of MFR is not enough as to detect the latter but just changes in viscosity highly dependent of the mixing degree, the molecular weight distribution, any degradation, et cetera. Note that the reaction between PR and SMM implies a new copolymers that may be better identified (if present) by DSC than from MFR. In any case, if the authors are convinced that the reaction really occurs, a mere FTIR spectra will be very probably  enough for such purposes.

There are many other concerns of the article, but the above mentioned are important enough as to support this reviewer decision.

In spite the interest of the topic. The article needs much amending that suggests to this reviewer recommending the REJECTION and further submission of the article.

Author Response

(The authors gave the same response as above.)

Round 2

Reviewer 2 Report

Equally than in  the previous revision draft this reviewer opines that the paper titled “ Physical and morphological properties of tough and transparent PMMA-based blends modified with polyrotaxane” appears as very interesting, showing as the main novelty the use of polyrotaxane as a polymer capable of improving the mechanical properties of the PMMA without much loss of transparency.  So, once that the authors have solved in a quite pretty way  a part of the concerns of this reviewer, but fail in three of them, my recommendation has now evolved from rejection and resubmission to Major revision. Below the authors comments the opinion of this reviewer is written in blue.

Point 1: Please, provide some properties of the polymers and the compatibilizer used and not just the supplier. This is important for traceability and easier comprehension of the further discussed.

Response 1: We have added the properties of the materials used (molecular weight) to improve the reproducibility of our experiments. The Tg, Tm, and MFR are also shown in Table 1.

Reviever Second Draft: Ok. The added by the authors may be enough.

Point 2: In the DSC study, please, provide the amount of sample used and the number of scans.

Concerning also to the DSC study, the authors have performed a single heating scan. Any reason for it? Note that this scan incorporates the information of both the processing step and of the material itself. So, this reviewer wonders if the authors have considered the latter. Since the processing conditions used are the same for all the samples in spite of the very different nature of the compounds (the authors used the same condition for processing so different neat materials as PMMA and SMM, and all the other compounds), and the effect of this may play a very different role depending on the concentration. Note that when a material is tested, the information obtained id a mixture of that provided by the processing operations plus the one due to the structure of the material.  So, this reviewer wonders why the authors have not performed a cooling and a further second heating scan in order of erasing the previous thermal history of the material, and then isolating the thermal behavior of the material itself.

Response 2: We have added detailed measurement conditions to the text (lines 119-121).

We ventured to use the first-scan thermogram of the DSC measurement in order to correlate the morphology and mechanical properties of the injection-molded parts. Another reason for this was the concern that repeated overheating of the specimen would accelerate the chemical reaction.

Reviewer Second Draft: Ok Convincing.

Point 3: Is there any reason for using TEM in absence of other observations at lower magnification levels? Since microscopy follows a hierarchical scheme, the logical is to observe from eye to TEM by using lower resolution techniques providing a more accurate panorama about the mixing and dispersion of the phases. This is critical since the authors recognize in the discussion section that some differences in properties may be due to a bad mixing, and so to non homogeneous samples.

Response 3: All the polymer blend materials prepared in this study showed high transparency. In particular, even PR10, which had the lowest transmittance, showed an average transmittance of 60%. Based on the results, no PR domains larger than a few microns were observed, and therefore, only high magnification TEM observations were performed.

Reviewer Second Draft: Ok, Concerns convincingly solved.

Point 4: In the Tensile section, the authors avoid to provide information about the use or not of extensometer. The gap distance is also needed. This is important to make opinion about the precision of the results reported.

Response 4: We have added information on experimental tensile testing to the text (lines 150-151).

Reviewer Second Draft: Ok. Concern solved convincingly.

Point 5: In the Impact section, this reviewer wonders if the authors have used the dumb-bell shaped specimens. This is not clear in the text. Additionally, the authors must inform if the samples have been flat-wise of edgewise used. The information about the champions for the impact testing must be better defined (for traceability sake). Also in this section, it must be remarked that just the mass of the hummer is not sufficient to define the test conditions by itself. At least, it is necessary to inform about the energy provided (dependent of the hammer angle) and the testing rate. In absence of these (jointly to the information about the specimens), the information is incomplete as to replicate the experiments. Please, provide the number of samples measured.

Response 5: The specimens for the impact test were of the dumbbell type. This information can be found in the line 156. More detailed experimental conditions for the Charpy impact test (hammer lift angle and blow point speed) have been added. In addition, the number of specimens measured has been noted (line 153-158).

Reviewer Second Draft: This is a point that is not totally solved. Flat-wise or edgewise?. None is said. Nevertheless, the use of dumbbell specimens is not a good choice in impact testing. What is the reason of using them directly? Why is the reason of using no notched specimens?

Point 6: The figure 5 needs error bars. Please, provide them. Based in my own experience on this type of tests. The error bars for the PR5, PR10, PSR5 and PSR6 may surely overlap, and so, any discussion on these results must be necessarily spurious.

Response 6: We added error bars to Figure 5; the error bars for PR5, PSR5, and PSR10 were duplicated, but PR10 showed a lower value.

Reviewer Second Draft: Ok. The latter may be acceptable.

Point 7: The Figure 7 includes error bars for impact parameter, but in this occasion, the error bars clearly overlap. Please, a discussion for this is needed. Nevertheless, in absence of the test conditions, sample better specifications and number of samples tested, the results hardly may be robust for a reader.

Response 7: The error bars for the values of the un-notched Charpy impact test in Figure 7 are large. This is due to the fact that the specimens do not have a notch, so the locations of stress concentration are varied and the impact values are also varied. Therefore, we do not discuss the differences between samples in the text. We only note in this section that PSR5 and PSR10 were non-destructive.

Reviewer Second Draft: the same as in point 5. What is the reason for using un-notched specimens while the dispersion of data increases enormously and then the data obtained gain in uncertainly? Provide a robust answer to the latter. The use of un-notched specimens implies the use of many more specimens to make the results being robust enough. In this scenario, the use of just 5 specimens is clearly insufficient. Notice that, even in the case of using notched specimens, the recommendations are usually to employ more than 10.

Point 8: The authors use the MFR to suggest that the reaction between SMM and PR may be taken place: ” The MFRs of PSR5 and PSR10 were significantly lower than that of PMSM, which suggested that the maleic anhydride groups in SMM reacted with hydroxy groups in PR to form chemical bonds between SMM and PR.”. Note that this is a bold suggestion in absence of more evidences. The sensibility of MFR is not enough as to detect the latter but just changes in viscosity highly dependent of the mixing degree, the molecular weight distribution, any degradation, et cetera. Note that the reaction between PR and SMM implies a new copolymers that may be better identified (if present) by DSC than from MFR. In any case, if the authors are convinced that the reaction really occurs, a mere FTIR spectra will be very probably  enough for such purposes.

Response 8: We discussed the possibility of the reaction between PR and SMM together with the amount of Tg and Tm shifted by DSC, as well as MFR. As a result, we believe that a PR-SMM copolymer was formed, leading to the shift in Tg.

Reviewer Second Draft: In Figure 2, a slight shift of the Tg may be observed, but this fact does not necessarily imply the existence of chemical bonds between PR and SMM. The mere change of the interaction scenario due to the presence of a third element as it is SMM is enough to produce this kind of displacement in the transition.To conclude that this phenomenon is due to a chemical bond that requires further experimentation. So, eliminate the latter of the text or make the additional experiments suggested in the previous revision draft.

So, in the light of the previous suggestions and taking into account the efforts by the authors, this reviewer would like to recommend a MAJOR revision on the basis of the concerns on points 5, 7 and 9, that need a profound amending.

Author Response

We appreciate the valuable comments from reviewers. We have corrected the ambiguous wording based on the reviewer's point. Please see the attached file.
